## Research Article

virtual reality; refugees; Uganda; depression; self-compassion; youth

**Corresponding author:**
Carmen H. Logie;
Email: carmen.logie@utoronto.ca

# Findings from the Tushirikiane-4-MH (supporting each other for mental health) mobile health–supported virtual reality randomized controlled trial among urban refugee youth in Kampala, Uganda

Carmen H. Logie[1,2,3,4] ⬥, Moses Okumu[5,6], Zerihun Admassu[1] ⬥,

Frannie MacKenzie[1], Lesley Gittings[7,8], Jean-Luc Kortenaar[9], Naimul Khan[10],

Robert Hakiza[11], Daniel Kibuuka Musoke[12], Aidah Nakitende[12], Brenda Katisi[11],

Peter Kyambadde[13,14], Richard Lester[15] and Lawrence Mbuagbaw[16,17,18,19,20,21]

[1]Factor-Inwentash Faculty of Social Work, University of Toronto, Toronto, Ontario, Canada; [2]Women's College Research Institute, Women's College Hospital, Toronto, Ontario, Canada; [3]United Nations University Institute for Water, Environment & Health, Hamilton, Ontario, Canada; [4]Centre for Gender and Sexuality Health Equity, Vancouver, British Columbia, Canada; [5]School of Social Work, University of Illinois Urbana-Champaign, USA; [6]Uganda Christian University, Mukono, Uganda; [7]School of Health Studies, Western University, London, Canada; [8]Centre for Social Science Research, University of Cape Town, South Africa; [9]Dalla Lana School of Public Health, University of Toronto, Toronto, Ontario, Canada; [10]Department of Electrical, Computer and Biomedical Engineering, Toronto Metropolitan University, Toronto, Ontario, Canada; [11]Young African Refugees for Integral Development (YARID), Kampala, Uganda; [12]International Research Consortium, Kampala, Uganda; [13]National AIDS and STI Control Programme, Ministry of Health, Kampala, Uganda; [14]Most At Risk Population Initiative, Mulago Hospital, Kampala, Uganda; [15]Department of Medicine, University of British Columbia, Vancouver, British Columbia, Canada; [16]Department of Health Research Methods, Evidence and Impact, McMaster University, Hamilton, Ontario, Canada; [17]Department of Anesthesia, McMaster University, Hamilton, ON, Canada; [18]Department of Pediatrics, McMaster University, Hamilton, ON, Canada; [19]Biostatistics Unit, Father Sean O'Sullivan Research Centre, St Joseph's Healthcare, Hamilton, ON, Canada; [20]Centre for Development of Best Practices in Health (CDBPH), Yaoundé Central Hospital, Yaoundé, Cameroon and [21]Division of Epidemiology and Biostatistics, Department of Global Health, Stellenbosch University, Cape Town, South Africa

## Abstract

Virtual reality (VR) for mental health promotion remains understudied in low-income humanitarian settings. We examined the effectiveness of VR in reducing depression with urban refugee youth in Kampala, Uganda. This randomized controlled trial assessed VR alone (Arm 1), VR followed by Group Problem Management Plus (GPM+) (Arm 2) and a control group (Arm 3), with a peer-driven and convenience sample of refugee youth aged 16–25 in Kampala. The primary outcome, depression, was measured with the Patient Health Questionnaire-9. Secondary outcomes included: mental health literacy, mental health stigma, self-compassion, mental well-being and adaptive coping. Analyses were conducted at three time points (baseline, 8 weeks, 16 weeks) using generalized estimating equations. Among participants ($n$ = 335, mean age: 20.77, standard deviation: 3.01; cisgender women: $n$ = 158, cisgender men: $n$ = 173, transgender women: $n$ = 4), we found no depression reductions for Arms 1 or 2 at 16 weeks compared with Arm 3. At 16 weeks, mental health literacy was significantly higher for Arm 2 compared with Arm 3, and self-compassion was significantly higher in Arm 1 and Arm 2 compared with Arm 3. VR alongside GPM+ may benefit self-compassion and MHL among urban refugee youth in Kampala, but these interventions were not effective in reducing depression.

## Impact statement

Youth living in low- and middle-income country (LMIC) humanitarian settings disproportionately experience mental health challenges. Despite virtual reality (VR) showing promise in reducing mental health challenges and promoting mental well-being in high-income settings, its potential benefits for mental health are understudied in LMIC at large, including humanitarian settings. To address this knowledge gap, we conducted a randomized controlled trial with a peer-driven and convenience sample of urban refugee youth in Kampala, Uganda, aged 16–25. This involved developing and evaluating the effectiveness of a VR intervention focused on mental health literacy, stigma and coping strategies. We tested the VR intervention alone, as well as followed by Group Problem Management Plus (GPM+), a World Health Organization group-based brief transdiagnostic psychological intervention developed for adults experiencing

adversity. We examined the effectiveness of VR alone (Arm 1) and VR followed by GPM+ (Arm 2), compared with a control arm (Arm 3), in reducing depression and improving secondary mental health outcomes (mental health literacy, mental health stigma, self-compassion, mental well-being and adaptive coping). We found no depression reductions for Arm 1 or Arm 2 at 16 weeks compared with Arm 3, and in fact we found higher depression among Arm 2 but in gender-disaggregated analyses, this was only significant among young men. At 16 weeks we found significantly higher mental health literacy for Arm 2 compared with Arm 3, and significantly higher self-compassion in Arm 1 and Arm 2, compared with Arm 3. These findings add to the limited evidence base of VR mental health interventions in LMIC, to signal that VR can benefit self-compassion among urban refugee youth in Kampala. We also show that alongside GPM+, VR can improve mental health literacy among this population. While VR shows promise in improving positive mental health outcomes, these strategies were not effective in reducing depression.

## Introduction

At the end of 2022, there were 112.6 million forcibly displaced people across the globe, 40% of whom were children and youth under the age of 18 (UNHCR 2022a). The impacts of forced displacement on the mental health of children and youth are profound. Mental health challenges disproportionately affect persons in humanitarian contexts due to multiple stressors including exposure to violence, trauma, food insecurity and social marginalization (Logie *et al.* 2020; Silove *et al.* 2017). Yet most forcibly displaced persons do not receive needed mental health support due to insufficient service availability, among other barriers (Silove *et al.* 2017).

Uganda is the largest refugee-hosting nation in Africa, with over 1.58 million refugees in 2022 (UNHCR 2022b). More than 111,000 of those refugees live in the city of Kampala, 27% of whom are youth aged 15–24 years (UNHCR 2022b). As the number of displaced persons rises globally, there is also an increasing trend of urbanization, with more than 60% of refugees and 80% of internally displaced persons living in urban settings (Muggah and Abdenur 2018). Urban refugees in low- and middle-income countries (LMICs) face unique challenges including exploitation, discrimination and language barriers that may present barriers to employment and in turn increase reliance on informal economies (Muggah and Abdenur 2018). Many urban refugees in LMICs, including those living in Kampala, live in informal settlements, such as slums, that experience heightened socio-environmental stressors (*e.g.*, violence, poverty) (Bukuluki *et al.* 2020; Saliba and Silver 2020; Women's Refugee Comission 2011). These daily stressors, as detailed in a study with urban Somali refugees in Kenya, may converge with histories of conflict-related trauma to exacerbate mental health challenges (Im *et al.* 2017). Prior research with a longitudinal cohort of urban refugee youth in Kampala noted a moderate to severe depression prevalence of 27.5%, with no significant differences before and after COVID-19, that was associated with food insecurity, experiencing violence, and lower social support (Logie *et al.* 2022).

Significant knowledge gaps remain in understanding effective approaches for improving mental health with adolescents and youth in humanitarian contexts, particularly regarding reducing depression and anxiety (Purgato *et al.* 2018). A recent systematic review and meta-analyses of mental health and psychosocial support programs with children and youth in LMIC humanitarian emergencies found cognitive behavioral therapy (CBT) was associated with reduced depression but reported inconsistent findings for other modalities (narrative exposure therapy, interpersonal and body psychotherapy, psychosocial programs, psychoeducation) (Bangpan *et al.* 2024). Authors in turn call for tailored programming to meet youth mental health needs in LMIC humanitarian contexts, particularly in Africa, which remain understudied relative to the number of refugees hosted (Bangpan *et al.* 2024). Strategies

with youth in Uganda have explored psychotherapeutic interventions and creative expressive techniques. For example, a study examining the efficacy of an Interpersonal Psychotherapy (IPT) intervention and an activity-based creative play intervention in Northern Uganda demonstrated a positive effect and significant improvement in depressive symptoms among girls in the IPT condition; however, the creative play intervention showed no effect on depressive severity (Bolton *et al.* 2007). Another study examining the effectiveness of sports-for-development programs in Uganda found positive effects for boys only (Richards *et al.* 2014). A randomized controlled trial (RCT) with youth former child soldiers aged 12–25 in Northern Uganda found narrative exposure therapy was associated with significant reductions in post-traumatic stress disorder (PTSD) symptom severity, functional impairment and guilt – but not stigma or depression (Ertl *et al.* 2011). Despite documented mental health challenges among urban refugee youth in LMIC contexts, there is a dearth of mental health interventions focused on urban refugee youth mental health in LMICs, including in Uganda (Saliba and Silver 2020). This reflects larger knowledge gaps regarding rural–urban migration and mental health in LMICs (Meyer *et al.* 2017), and other health issues among urban refugees in LMICs (Logie *et al.* 2024b).

This study aims to address knowledge gaps regarding efficacious interventions to reduce depression among urban refugee youth in Kampala, Uganda, by introducing interventions that are novel to this population and tailored to their unique mental health needs. We evaluated the effectiveness of a virtual reality (VR) experience focused on mental health literacy and psychological first aid skills (World Health Organization *et al.* 2011) implemented on its own, and this VR experience followed by Group Problem Management Plus (GPM+), a World Health Organization (WHO) group-based brief transdiagnostic psychological intervention developed for adults experiencing adversity (World Health Organization 2020). The primary study objective is to determine the effectiveness of VR compared to the standard of care (SOC), and VR followed by GPM + compared with the SOC, in reducing depression. Secondary objectives include examining the effectiveness of these two intervention approaches on: (1) increasing mental health literacy, (2) reducing mental health stigma, (3) increasing self-compassion, (4) increasing mental well-being and (5) increasing adaptive coping strategies, compared with the SOC.

## Background on intervention approaches

### *Virtual reality*

In high-income contexts, studies have pointed to the potential of VR for improving various mental health outcomes. VR-based technology allows users to experience an interactive three-dimensional environment where psychotherapeutic interventions

such as CBT can be applied (Rowland *et al.* 2022). A systematic review of VR treatment for PTSD among adults found it was more effective than a control group and as effective as other therapeutic modalities; however, the small number of studies and low study quality underscore the need for additional research (Eshuis *et al.* 2021). Scoping review findings of VR for treating depression and anxiety with CBT approaches reported reduced anxiety or depression symptoms, but few studies used an RCT design (Baghaei *et al.* 2021). Another systematic review examining the efficacy of VR interventions for emotional disorders reported that most VR studies were effective compared to waitlist and control conditions in reducing self-reported social anxiety, panic disorder, PTSD; however, there was heterogeneity in findings (Rowland *et al.* 2022). Despite these promising findings, most VR studies were focused on adults in high-income settings, revealing knowledge gaps of their efficacy with youth in LMIC and/or humanitarian contexts.

### Group problem management plus

GPM+ is a group-based brief psychological intervention to help adults impaired by distress in communities affected by adversity (World Health Organization 2020). This program is an adaptation of Problem Management Plus (PM+), an individual intervention for adults affected by adversity (Dawson *et al.* 2015). GPM+ programs were designed as scalable group-based brief psychological interventions that can be delivered by nonspecialists in LMIC where mental health infrastructure is limited or unable to meet the psychosocial needs of the population (Sangraula *et al.* 2020). PM+ has been shown to reduce psychological distress among women in urban Kenya (Bryant *et al.* 2017) and indicates reductions on depression, anxiety, PTSD symptoms and self-identified problems among Syrian refuges in the Netherlands (Graaff *et al.* 2023). The group-based delivery format, GPM+, was feasible and acceptable among refugee and conflict-affected individuals in Jordan (Akhtar *et al.* 2020), Pakistan (Khan *et al.* 2017) and Nepal (Sangraula *et al.* 2020). A study examining the efficacy of GPM+ among conflict-affected adults in Nepal found modest reductions in psychological distress and depression symptoms (Jordans *et al.* 2021). The WHO states this program is likely efficacious among adolescents aged 16 years and older (World Health Organization 2020); however, GPM+ has yet to be evaluated with refugee adolescents and youth. As many VR mental health interventions integrate CBT and other evidence-based approaches (Rowland *et al.* 2022), we also tested VR followed by in-person GPM+ to explore any additional added value by combining these approaches.

## Methods

### Study design and setting

Tushirikiane (roughly translating to "Supporting Each Other" in Swahili) for Mental Health (Tushirikiane-4-MH) was a three-arm RCT conducted in five informal settlements in Kampala, Uganda. We evaluated the impact of virtual reality (VR) alone, and VR followed by GPM+, in comparison with a control group, on *primary* (depression) and *secondary* (mental health literacy, mental health stigma, self-compassion, mental well-being, adaptive coping) outcomes among refugee youth in Kampala. Data were collected at three time points: before the intervention implementation, 8 weeks following the intervention and 16 weeks following the intervention. Full details regarding the trial and the study setting

have been described elsewhere (Logie *et al.* 2021a); the trial is registered at ClinicalTrials.gov (NCT05187689).

### Participants and recruitment

Participants were recruited from the Tushirikiane HIV self-testing cohort study, whereby participants aged 16–24 years old were recruited between 2020 and 2021 (Logie *et al.* 2021a); the Tushirikiane cohort was continued for implementing a COVID-19 prevention study (Logie *et al.* 2021b). In 2022, the cohort participants were invited to participate in the present study (Tushirikiane-4-MH) by peer navigators (PNs); to reach the desired sample size, we conducted additional purposive recruitment. PN purposively recruited 16- and 17-year-old participants to refresh the cohort as this age range was no longer present in the existing cohort. Baseline data were collected in April 2022, the intervention was conducted in June–July 2022, 8-week follow-up surveys were conducted in August–September 2022 and 16-week follow-up surveys were conducted in November–December 2022. Participants were recruited using peer-driven and convenience sampling methods from five informal settlements in Kampala where a large proportion of refugees live; informal settlements were grouped into three study sites based on geographical proximity (1: Kabalagala and Kansanga, 2: Katwe and Nsambya, 3: Rubaga). After recruitment was complete, each location was randomly assigned to one of the three study arms. PNs, who identify as refugee or displaced persons aged 18–25 years, were engaged to help with participant recruitment and retention. PNs were supervised by a study coordinator at YARID and trained by the PI on the study design, interventions, ethics and psychological first aid over multiple training sessions.

Eligibility criteria included: (1) currently living in one of the five selected informal settlements in Kampala (Kabalagala, Kansanga, Katwe, Nsambya or Rubaga); (2) identifying as a refugee or displaced person, or having refugee or displaced parents; (3) aged 16–25 years; (4) owning or have daily access to a mobile phone; and (5) speaking French, English, Kirundi, Kinyarwanda or Swahili. Interested participants were screened for eligibility by a trained PN by phone, in person or WhatsApp.

### Intervention implementation

The study was designed as a three-arm RCT, consisting of two treatment arms and one control group randomized at a 1:1:1 ratio by the study coordinator using an online random number generator: Arm 1 was the VR experience, Arm 2 was the VR experience followed by GPM+ and Arm 3 was the control arm. Details of the study have been described elsewhere (Logie *et al.* 2021a). We briefly describe the intervention below:

#### Arm 1: VR

Participants in this arm received a single 15-min immersive and interactive VR session in a private room or in an outdoor setting. The VR experience was developed to equip participants with information and tools to improve their mental health outcomes. Components of the VR experience included three separate scenarios that included an interaction with different characters whereby psychosocial information was shared: the first scenario discussed depression symptoms and aimed to improve mental health literacy and psychological first aid skills; the second scenario included descriptions of a character's lived experience of mental health stigma and isolation and aimed to reduce mental health stigma; and the third scenario involved teaching and practicing self-compassion and emotional regulation exercises.

To ensure feasibility and sustainability of VR in this setting, PN co-developed the experience ensuring it was both simple and relevant for use with urban refugee youth. As illustrated in this screenshot from the VR experience, participants were guided to visit the three characters in the scene who were identified with a speech bubble above them (Supplementary Figure 1).

The PN facilitated the use of the VR with each participant, describing what VR is, how it will work, what the participant can expect and provided instruction in using the hand controller to move around the scene and meet the three characters. After extensive pilot testing with the PNs, the length of the VR was set to 15 min, and the interaction was minimized so the participant used hand controllers to move around the scene and touch the characters with speech bubbles, which would then trigger a prerecorded psychoeducation session as described above.

### Arm 2: VR and GPM+

Participants in this arm participated in the VR intervention (described above) as well as GPM+. The GPM+ intervention consisted of a manualized five-session, 3-h/session, led by two trained PNs.

### Arm 3: Control

Participants in this arm received a list of mental health resources in Kampala.

The Arm 1 and Arm 2 intervention arms received weekly SMS two-way check-ins and SMS mental health awareness messages in their preferred language on the WelTel platform (Lester *et al.* 2010). Any participant who expressed discomfort, mental health concerns or requested assistance was referred to YARID's social workers and their PN in 48 h. To provide additional group-based support to participants, Arm 1 and Arm 2 participants were invited to take part in weekly WhatsApp group discussions with PN. All study members, including those in the control group, had access to mental health support from YARID resources and trained social workers as needed.

### Outcomes

The primary outcome was depression assessed using the Patient Health Questionnaire-9 (PHQ-9; Cronbach's alpha = 0.82). Responses for the nine items were summed up with the total scores ranging from 0 to 27, with higher scores indicating higher levels of depression (Negeri *et al.* 2021).

Secondary outcomes included **mental health literacy** which was assessed with the 16-item short version of the Mental Health Literacy Scale (Campos *et al.* 2022). Higher scores indicate higher levels of mental health literacy (Cronbach's alpha = 0.81). **Mental health stigma** was measured by the 7-item Day's Mental Illness Stigma Scale (Day *et al.* 2007). The higher the score, the more negative the attitude toward people with mental illness (Cronbach's alpha = 0.81). **Self-compassion** was measured using items from the youth version of the Self-Compassion Scale (Neff *et al.* 2021). For this study, 6 of the 17 items of the SCS-Y, from the subscales of Self-Kindness, Shared Humanity, Isolation and Overidentification, were used. Higher scores indicate higher levels of self-compassion (Cronbach's alpha = 0.77). **Mental well-being** was measured using the WHO-5 Wellbeing Index. The total scale ranges from 0 (worst possible quality of life) to 25 (best possible quality of life) (Bech *et al.* 2003). The scale reliability coefficient was 0.88. **Adaptive coping** (Cronbach's alpha = 0.73) was assessed with KidCOPE, a brief

checklist to measure cognitive and behavioral coping in children and adolescents (Spirito *et al.* 1988).

### Sample size calculation

Our pretrial power calculation indicated that 330 participants, with 110 per study arm, were required to detect a difference of 3 points in mean depression score (moderate effect size). This assumes an intraclass correlation of 0.01 and standard deviation (SD) of 7, at a 5% level of significance with 80% power and anticipation of a 10% attrition rate. Calculations were performed using RStudio version 3.3.0, based on the formula for multiple comparisons of proportions and adjusted for design effects (Chow *et al.* 2007).

### Data collection and management

Participants' data were collected at three time points from all study arms: pre-intervention, 8-week post-intervention and 16-week post-intervention follow-up. Standardized questionnaires were administered by trained research assistants using SurveyCTO, a secured tablet-based application (Dobility, Cambridge, USA). To maintain confidentiality, all participants were given a unique case ID, and no personal identifying information was collected in the survey.

### Statistical analysis

All primary and secondary analyses were based on modified intention-to-treat principles (Montedori *et al.* 2011) in accordance with the consolidated standards of reporting trials (CONSORT) statement (Abraha and Montedori 2010) using complete case analysis, as our missing data were less than 10%, which is generally acceptable for maintaining the validity and power of the study.

First, we compared the study arms' demographic and other baseline characteristics for any differences across study arms using analysis of variance for continuous variables and chi-square tests for categorical variables. Characteristics of study completers *versus* those lost to follow-up were compared using independent *t*-tests for continuous variables and chi-squared ($\chi^2$) tests for categorical variables, and no significant differences were found.

The primary analysis comparing the three study arms at the three assessment time points was conducted by a population average model using generalized estimating equations (GEEs; Liang and Zeger 1993). For each outcome of interest, we used an unstructured correlation matrix to estimate the intervention effect across time while adjusting for potential confounding factors. We used robust standard errors to account for clustering. To specify a GEE model, we used a Poisson distribution and an identity link function for all response variables. GEEs account for correlated data due to multiple assessments of individual participants in longitudinal study designs (Liang and Zeger 1986). GEE models estimated the effects of (1) "Time" (time 1, time 2, time 3), (2) "Study Arm" (VR, VR + GPM, Control) and (3) the "Time × Study Arm" interaction. p-values associated with the interaction term were used to determine the statistical significance of any differences between the study arms at each time point. In these models, the main coefficient of the interaction term reveals the mean difference between intervention and control arms, controlling for baseline differences and secular trends. Each model was first conducted without adjustment, then with adjustment for age and gender, which were specified as *a priori*, as well as characteristics with baseline imbalances between

study arms including the baseline study outcome scores for each respective model.

To explore gender differences in primary and secondary intervention outcomes, which is recommended in global health research due to the ways in which sociocultural norms shape gender expectations and roles (Shapiro *et al.* 2021), we conducted gender stratified analyses (*i.e.*, men and women separately). Intervention effects are expressed as crude $(\beta)$ and adjusted coefficient (a $\beta$), along with 95% confidence intervals (CIs). All regression analyses were performed as a complete case analysis and employed 2-tailed tests of significance and α = 0.05. Data were analyzed using Stata 14.2 (StataCorp, College Station, TX).

Consistency between the results of the primary analysis and the results of our sensitivity analysis was investigated to examine intervention effects by participating in any intervention arm (VR alone, VR followed by GPM+) *vs.* the control group (Thabane *et al.* 2013).

## Results

### Enrolment and baseline characteristics

A total of 335 eligible and consenting participants were randomly assigned to VR (Arm 1) (*n* = 113), VR and GPM+ (Arm 2) (*n* = 112) and the control (SOC) (Arm 3) (*n* = 110) based on living in one of the three study sites randomized to each arm. The distribution of participant sociodemographic characteristics at baseline is reported in Table 1.

The mean age and SD of respondents was 20.8 years (SD: 3.01); nearly half (47.2%) were cisgender women. Most participants (75.5%) were from the Democratic Republic of the Congo. Baseline demographic characteristics were largely balanced across study arms, apart from age, place of birth, education level and food insecurity status (Table 1).

Participants' baseline depression score was 7.23 (SD = 5.54). There was a statistically significant difference in baseline depression scores between the study arms (p < 0.001), with Arm 3 having the highest depression baseline scores (mean = 9.22; SD = 6.57), followed by Arm 2 (mean = 6.91; SD = 4.70), and then Arm 1 (VR alone) (mean = 5.62; SD = 3.80). The mean baseline mental health literacy scores differed significantly across study arms (p < 0.001), 54.87 (SD = 7.17) for Arm 1, 48.53 (SD = 7.29) for Arm 2 and GPM+, and 51.68 (SD = 5.82) for Arm 3 (Table 2). Baseline mental health stigma scores also differed significantly across study arms (p < 0.001). Control arm participants had the highest stigma scores (32.42, SD = 4.17), while the VR and VR + GPM Arms showed similar scores of 29.59 (SD = 7.02) and 29.46 (SD = 5.22), respectively. These baseline differences were adjusted for in multivariable analyses. Participant's mean scores for baseline and post-intervention (8 weeks, 16 weeks) outcome characteristics are presented in Table 2.

### Study participant flow

For the 335 participants enrolled, the numbers and percentages of retained participants at the 8- and 16-week time points are shown in the CONSORT flow chart (Supplementary Figure 2). Excellent retention was found with 95.8%, and 94.8%, of participants retained at the 8-week follow-up, and 16-week follow-up, respectively. Data were missing for less than 5% of participants across all variables and time points and can thus be considered low. Sociodemographic characteristics were compared between those who had completed and those who had not completed the study at each time point, and

there were no statistically significant differences found. The final data analysis was thus restricted to the participants who remained in the study.

### Intervention effects on primary and secondary outcomes

Tables 3 and 4 show study arm comparisons on depression, mental health literacy, mental health stigma, self-compassion, *mental well-being* and *adaptive coping strategies* (main effects: interaction between group membership × time of measurement) measured at 8 weeks and 16 weeks. Multivariable analysis was conducted, after adjusting for prespecified covariates (age, gender) and baseline imbalances (place of birth, level of education, food security and relationship status). For gender-stratified analyses, we adjusted for age and baseline imbalances in place of birth, level of education, food security and relationship status.

#### Depression

Participants in Arm 1 (aβ = 0.27, 95% CI = −1.05, 1.58; p = 0.691) and Arm 2 (aβ = −0.36, 95% CI = −1.92, 1.20; p = 0.650) did not show any significant changes in depression scores when compared to the control arm at 8-weeks. At 16 weeks, Arm 2 participants reported statistically significant higher odds of depression (aβ = 2.24, 95% CI = 0.54, 3.93; p = 0.010) compared to the control arm; this association was significant among men (aβ = 3.67, 95% CI = 0.95, 6.38; p = 0.008), but not women. There was no significant difference in depression scores at 16 weeks in Arm 1 compared to Arm 3.

Mental health literacy. Adjusted regression scores for mental health literacy indicate that participants in both intervention arms had statistically significant higher mental health literacy compared to the control arm at 8 weeks: Arm 1 (aβ = 3.07, 95% CI = 1.09, 5.05; p = 0.002), and Arm 2 (aβ = 5.62, 95% CI = 3.61, 7.64; p < 0.001). At 16 weeks, participants in Arm 2 had significantly higher mental health literacy compared to the control arm (aβ = 2.98, 95% CI = 0.69, 5.26; p = 0.011); there were no significant differences for Arm 1 participants (aβ = 0.63, 95% CI = −1.21, 2.47; p = 0.502) compared to the control arm.

Mental health literacy scores among young men in Arm 2 were significantly higher than the control arm at 8 weeks (aβ = 7.35, 95% CI = 4.67, 10.03; p < 0.001) and 16 weeks (aβ = 3.25, 95% CI = −0.13, 6.63; p = 0.059). At 8 weeks, young men in Arm 1 (VR alone) reported higher mental health literacy scores compared to the control arm (aβ = 4.30, 95% CI = 1.58, 7.02; p = 0.002), but this was not sustained at 16 weeks (p = 0.567). Young women participants in Arm 2 reported significantly higher mental health literacy at 8 weeks compared to the control arm (aβ = 3.94, 95% CI = 0.95, 6.93; p = 0.010), but this was not sustained at 16 weeks. No significant changes were seen in mental health literacy scores among young women in Arm 1 compared to the control group.

Mental health stigma. There were no significant differences in mental health stigma at 8 weeks between Arm 1 or Arm 2 compared to the control arm, and there were no significant differences between Arm 2 and the control arm at 16 weeks. However, at 16 weeks, participants in Arm 1 reported higher mental health stigma compared with the control arm (aβ = 4.25, 95% CI = 1.92, 6.57; p < 0.001). In gender-stratified analyses, the only statistically significant mental health stigma differences at 8 weeks were among young men in Arm 1 who reported lower mental health stigma compared to the control arm (aβ = −3.10, 95% CI = −5.97, −0.23; p = 0.034). At 16 weeks, there were no significant differences in

**Table 1.** Baseline characteristics of refugee youth participants enrolled in the Tushirikiane-4-MH study, Kampala, Uganda, 2022

| Demographic characteristics | Total N = 335 | VR (alone) – Arm 1 N = 113 | VR and GPM+ – Arm 2 N = 112 | Control –Arm 3 N = 110 | p-Value |
|---|---|---|---|---|---|
| Age, mean (SD) | 20.77(3.01) | 20.14(3.03) | 20.90(2.85) | 21.27(3.06) | 0.016 |
| Gender, N (%) | | | | | 0.188 |
| Woman (cisgender) | 158(47.16) | 48(42.48) | 60(53.57) | 50(45.45) | |
| Man (cisgender) | 173(51.64) | 64(56.64) | 52(46.43) | 57(51.82) | |
| Transgender woman | 4(1.19) | 1(0.88) | 0 | 3(2.73) | |
| Place of birth, N (%) | | | | | < 0.001 |
| Democratic Republic of Congo | 253(75.52) | 104(92.04) | 95(84.82) | 54(49.09) | |
| Burundi | 34(10.15) | 0 | 2(1.79) | 32(29.09) | |
| Uganda | 14(4.18) | 5(4.42) | 8(7.14) | 1(0.91) | |
| Others | 34(10.15) | 4(3.54) | 7(6.25) | 23(20.91) | |
| Length of time in Uganda, N (%) | | | | | 0.079 |
| <5 years | 107(31.94) | 44(38.94) | 37(33.04) | 26(23.64) | |
| 6–10 years | 142(42.39) | 48(42.48) | 43(38.39) | 51(46.36) | |
| >10 years | 86(25.67) | 21(18.58) | 32(28.57) | 33(30.00) | |
| Employment status [n = 28] | | | | | 0.088 |
| No employment | 147(47.88) | 54(47.79) | 39(41.49) | 54(54.00) | |
| Student | 91(29.64) | 39(34.51) | 32(34.04) | 20(20.00) | |
| Employed (paid/unpaid) | 69(22.48) | 20(17.70) | 23(24.47) | 26(26.00) | |
| Highest level of education [n = 12] | | | | | <0.001 |
| Less than secondary | 150(46.44) | 64(57.66) | 66(61.68) | 20(19.05) | |
| Some secondary | 131(40.56) | 37(33.33) | 31(28.97) | 63(60.00) | |
| Secondary + | 42(13.00) | 10(9.01) | 10(9.35) | 22(20.95) | |
| Relationship status [n = 1] | | | | | 0.017 |
| No current partner | 133(39.82) | 53(46.90) | 36(32.43) | 44(40.00) | |
| Dating one partner/married | 154(46.11) | 50(44.25) | 61(54.95) | 43(39.09) | |
| Casual dating/multiple partners | 47(14.07) | 10(8.85) | 14(12.61) | 23(20.91) | |
| Have children [n = 1] | | | | | 0.491 |
| No | 298(89.22) | 102(90.27) | 101(90.99) | 95(86.36) | |
| Yes | 36(10.78) | 11(9.73) | 10(9.01) | 15(13.64) | |
| Food security [n = 3] | | | | | 0.005 |
| Food secure | 27(8.13) | 8(7.14) | 3(2.73) | 16(14.55) | |
| Food insecure | 305(91.87) | 104(92.86) | 107(97.27) | 94(85.45) | |

Note: SD, standard deviation; n-missing case; VR arm = Kansanga/Kabalagala settlement; VR & GPM+ = Nsambya/Katwe settlement; Control = Rubaga settlement.

mental health stigma among young men between study arms, but mental health stigma was higher among young women in Arm 1 (aβ = 5.77, 95% CI = 2.40, 9.13; p = 0.001) and Arm 2 (aβ = 3.17, 95% CI = 0.22, 6.12; p = 0.035) compared to the control arm.

**Self-compassion.** Compared with participants in the control arm, participants in both intervention arms showed significantly higher self-compassion at nearly all follow-up periods. At 8 weeks, the higher self-compassion scores in Arm 1 compared to the control group were not statistically significant (aβ = 1.34, 95% CI = −0.16, 2.85; p = 0.081), yet were significantly higher at 16 weeks (aβ = 2.85, 95% CI = 1.24, 4.46; p = 0.001). Self-compassion scores in Arm 2

were significantly higher than the control arm at 8 weeks (aβ = 1.90, 95% CI = 0.55, 3.25; p = 0.006) and 16 weeks (aβ = 3.36, 95% CI = 1.76, 4.95; p < 0.001). Findings at 16 weeks for both intervention arms showed self-compassion was higher among both young men and young women compared with the control arm.

### Mental well-being and adaptive coping
There were no statistically significant differences in mental well-being or adaptive coping between participants in the intervention versus control arms, except for at 8 weeks; Arm 2 reported lower adaptive coping than the control arm (aβ = −1.98, 95% CI = −3.49, −0.47; p = 0.010) but this difference was not sustained at 16 weeks.

**Table 2.** Distribution of mental health outcomes by intervention group and time point among refugee youth participants enrolled in the Tushirikiane-4-MH study in Kampala, Uganda, 2022

| Variables | Total | | VR (alone) | | VR and GPM+ | | Control | | p-Value |
|---|---|---|---|---|---|---|---|---|---|
| | N | Mean (SD) | N | Mean (SD) | N | Mean (SD) | N | Mean (SD) | |
| **Primary outcome** | | | | | | | | | |
| *Depression (PHQ–9)* | | | | | | | | | |
| Baseline | 324 | 7.23(5.42) | 112 | 5.62(3.80) | 104 | 6.91(4.70) | 108 | 9.22(6.75) | < 0.001 |
| 8 weeks | 312 | 6.89(5.88) | 100 | 5.70(3.99) | 110 | 6.03(4.16) | 102 | 8.99(8.09) | |
| 16 weeks | 302 | 6.87(5.69) | 100 | 5.31(4.36) | 103 | 7.32(5.54) | 99 | 7.97(6.67) | |
| **Secondary outcomes** | | | | | | | | | |
| *Mental health literacy* | | | | | | | | | |
| Baseline | 327 | 51.72(7.24) | 110 | 54.87(7.17) | 107 | 48.53(7.29) | 110 | 51.68(5.82) | < 0.001 |
| 8 weeks | 311 | 54.02(7.11) | 99 | 58.42(6.98) | 109 | 52.98(6.39) | 103 | 50.89(5.83) | |
| 16 weeks | 301 | 53.81(6.50) | 100 | 57.13(6.54) | 105 | 52.19(5.55) | 96 | 52.12(6.15) | |
| *Mental health stigma* | | | | | | | | | |
| Baseline | 332 | 30.46(5.76) | 112 | 29.59(7.02) | 112 | 29.46(5.22) | 108 | 32.42(4.17) | < 0.001 |
| 8 weeks | 312 | 29.55(7.21) | 99 | 27.95(8.07) | 110 | 28.04(6.93) | 103 | 32.72(5.44) | |
| 16 weeks | 304 | 29.87(7.18) | 100 | 30.77(7.05) | 106 | 28.82(7.09) | 98 | 30.07(7.34) | |
| *Self-compassion* | | | | | | | | | |
| Baseline | 331 | 20.28(4.08) | 112 | 21.12(4.63) | 109 | 19.57(3.82) | 110 | 20.13(3.59) | 0.0164 |
| 8 weeks | 312 | 21.26(4.09) | 99 | 22.06(4.96) | 110 | 21.46(3.48) | 103 | 20.26(3.58) | |
| 16 weeks | 306 | 21.36(4.99) | 100 | 22.57(5.25) | 107 | 22.09 (4.63) | 99 | 19.33(4.53) | |
| *Mental well-being* | | | | | | | | | |
| Baseline | 333 | 10.31(5.49) | 113 | 9.42(6.59) | 112 | 10.36(4.19) | 108 | 11.19(5.29) | 0.0574 |
| 8 weeks | 312 | 11.00(5.56) | 100 | 10.42(6.59) | 110 | 9.96(5.21) | 102 | 12.69(4.36) | |
| 16 weeks | 307 | 12.58(5.86) | 100 | 11.05(6.69) | 108 | 13.06(5.94) | 99 | 13.62(4.45) | |
| *Adaptive coping* | | | | | | | | | |
| Baseline | 324 | 7.70 (4.34) | 113 | 7.27 (4.06) | 102 | 6.99 (4.80) | 109 | 8.82 (3.96) | 0.0038 |
| 8 weeks | 307 | 8.04 (4.23) | 100 | 8.11 (4.04) | 105 | 6.39 (4.18) | 102 | 9.68 (3.84) | |
| 16 weeks | 304 | 7.53 (4.41) | 99 | 7.58 (3.74) | 106 | 7.02 (4.12) | 99 | 8.03 (5.24) | |

Note: SD, standard deviation; PHQ-9, Patient Heath Questionnaire; VR arm = Kansanga/Kabalagala settlement; VR & GPM+ = Nsambya/Katwe settlement; Control = Rubaga settlement. Prob > F (p-value) to indicate whether there are significant baseline differences among group means.

There were no significant differences in gender-stratified analyses at 16 weeks between Arm 1 or Arm 2 compared with the control arm in mental well-being or adaptive coping.

## Sensitivity Analysis

Participants who received any intervention at any time point exhibited significantly lower depression (a$\beta$ = −1.93, 95% CI: −3.20, −0.65; p = 0.003) and reduced mental health stigma (a$\beta$ = −1.03, 95% CI: −3.95, −1.90; p = 0.022) compared to those who did not receive any intervention. Additionally, participants who received interventions reported significantly higher mental health literacy (a$\beta$ = 1.90, 95% CI: 0.05, 3.29; p = 0.008), greater self-compassion (a$\beta$ = 1.74, 95% CI: 0.70, 2.79, p = 0.001) and higher adaptive coping (a$\beta$ = 1.42, 95% CI: 0.44, 2.39; p = 0.004) compared to those who did not receive any intervention (Table 5).

## Discussion

This study examined the effectiveness of the Tushirikiane-4-MH VR intervention with urban refugee youth in Kampala, Uganda, and findings highlight the benefits of VR in improving self-compassion and mental health literacy. However, we found no significant difference in the VR intervention arms in reducing depression, our primary outcome. This study adds to the very limited evidence base of the effectiveness of VR mental health interventions in LMIC humanitarian settings, and signals that VR can benefit positive mental psychology outcomes.

Our findings corroborate prior research on the potential benefits of VR on positive psychology outcomes, such as self-compassion, in high-income contexts (Li Pira *et al.* 2023). We found improved self-compassion among VR participants – referring to viewing oneself with kindness, awareness of common humanity and mindfulness of negative self-perception – which is a protective factor associated with resilience (Neff 2003, 2011). Studies reveal that VR

**Table 3.** Effectiveness of virtual reality intervention approaches on mental health outcomes among refugee youth participants in the Tushirikiane-4-MH study in Kampala, Uganda, 2022

| Variables | β | 95% CI | p-value | aβ* | 95% CI | p-value |
|---|---|---|---|---|---|---|
| **Depression (PHQ–9)** | | | | | | |
| Intervention effects at 8 weeks | | | | | | |
| VR (Arm 1) *vs.* control (Arm 3) | 0.29 | −0.99, 1.58 | 0.653 | 0.27 | −1.05, 1.58 | 0.691 |
| VR and GPM+ (Arm 2) *vs.* Control (Arm 3) | −0.58 | −2.12, 0.96 | 0.461 | −0.36 | −1.92, 1.20 | 0.650 |
| Intervention effects at 16 weeks | | | | | | |
| VR (Arm 1) *vs.* control (Arm 3) | 1.04 | −0.25, 2.33 | 0.114 | 0.93 | −0.40, 2.26 | 0.171 |
| VR and GPM+ (Arm 2) *vs.* control (Arm 3) | 1.94 | 0.28, 3.61 | 0.022 | 2.24 | 0.54, 3.93 | 0.010 |
| *Mental health literacy* | | | | | | |
| Intervention effects at 8 weeks | | | | | | |
| VR (Arm 1) *vs.* control (Arm 3) | 2.99 | 1.04, 4.95 | 0.003 | 3.07 | 1.09, 5.05 | 0.002 |
| VR and GPM+ (Arm 2) *vs.* control (Arm 3) | 5.35 | 3.38, 7.32 | <0.001 | 5.62 | 3.61, 7.64 | <0.001 |
| Intervention effects at 16 weeks | | | | | | |
| VR (Arm 1) *vs.* control (Arm 3) | 0.53 | −1.30, 2.36 | 0.571 | 0.63 | −1.21, 2.47 | 0.502 |
| VR and GPM+ (Arm 2) *vs.* control (Arm 3) | 3.23 | 1.06, 5.39 | 0.003 | 2.98 | 0.69, 5.26 | 0.011 |
| Mental health stigma | | | | | | |
| Intervention effects at 8 weeks | | | | | | |
| VR (Arm 1) *vs.* control (Arm 3) | −1.56 | −3.72, 0.59 | 0.156 | −1.73 | −3.92, 0.47 | 0.123 |
| VR and GPM+ (Arm 2) *vs.* control (Arm 3) | −1.15 | −3.29, 0.98 | 0.291 | −1.38 | −3.60, 0.84 | 0.223 |
| Intervention effects at 16 weeks | | | | | | |
| VR (Arm 1) *vs.* control (Arm 3) | 4.38 | 2.10, 6.65 | <0.001 | 4.25 | 1.92, 6.57 | <0.001 |
| VR and GPM+ (Arm 2) *vs.* control (Arm 3) | 1.96 | −0.24, 4.16 | 0.081 | 1.68 | −0.53, 3.89 | 0.136 |
| Self-compassion | | | | | | |
| Intervention effects at 8 weeks | | | | | | |
| VR (Arm 1) *vs.* control (Arm 3) | 1.20 | −0.28, 2.68 | 0.113 | 1.34 | −0.16, 2.85 | 0.081 |
| VR and GPM+ (Arm 2) *vs.* control (Arm 3) | 1.89 | 0.59, 3.19 | 0.004 | 1.90 | 0.55, 3.25 | 0.006 |
| Intervention effects at 16 weeks | | | | | | |
| VR (Arm 1) *vs.* control (Arm 3) | 2.58 | 0.99, 4.16 | 0.001 | 2.85 | 1.24, 4.46 | 0.001 |
| VR and GPM+ (Arm 2) *vs.* control (Arm 3) | 3.59 | 2.03, 5.15 | < 0.001 | 3.36 | 1.76, 4.95 | < 0.001 |
| Mental well-being | | | | | | |
| Intervention effects at 8 weeks | | | | | | |
| VR (Arm 1) *vs.* control (Arm 3) | −0.64 | −2.65, 1.38 | 0.537 | −0.68 | −2.73, 1.37 | 0.517 |
| VR and GPM+ (Arm 2) *vs.* control (Arm 3) | −1.68 | −3.46, 0.10 | 0.064 | −1.60 | −3.47, 0.28 | 0.095 |
| Intervention effects at 16 weeks | | | | | | |
| VR (Arm 1) *vs.* control (Arm 3) | −0.93 | −3.06, 1.19 | 0.389 | −1.07 | −3.24, 1.11 | 0.336 |
| VR and GPM+ (Arm 2) *vs.* control (Arm 3) | 0.52 | −1.46, 2.49 | 0.608 | 0.01 | −2.04, 2.06 | 0.990 |
| Adaptive coping | | | | | | |
| Intervention effects at 8 weeks | | | | | | |
| VR (Arm 1) *vs.* control (Arm 3) | −0.42 | −1.82, 0.99 | 0.561 | −0.64 | −2.05, 0.78 | 0.378 |
| VR and GPM+ (Arm 2) *vs.* control (Arm 3) | −1.69 | −3.17, −0.22 | 0.024 | −1.98 | −3.49, −0.47 | 0.010 |
| Intervention effects at 16 weeks | | | | | | |
| VR (Arm 1) *vs.* control (Arm 3) | 0.84 | −0.57, 2.26 | 0.242 | 0.57 | −0.81, 1.94 | 0.418 |
| VR and GPM+ (Arm 2) *vs.* control (Arm 3) | 0.44 | −1.19, 2.06 | 0.596 | −0.07 | −1.69, 1.55 | 0.932 |

Note: CI, confidence interval; PHQ-9, Patient Health Questionnaire; Intervention effect is estimated as the interaction between intervention arm and time point, calculated using generalized estimating equation linear regression models with an unstructured correlation matrix; aβ*. Adjusted for prespecified covariates (age, gender) and baseline imbalances (place of birth, level of education, food security and relationship status), and baseline outcome scores.

**Table 4.** Effectiveness of virtual reality intervention approaches on mental health outcomes among refugee youth participants in the Tushirikiane-4-MH study in Kampala, Uganda, 2022, stratified by gender

| Variables | Young men | | | Young women | | |
|---|---|---|---|---|---|---|
| | aβ* | 95% CI | p-value | aβ* | 95% CI | p-value |
| Depression (PHQ–9) | | | | | | |
| Intervention effects at 8 weeks | | | | | | |
| VR (Arm 1) *vs.* control (Arm 3) | 0.76 | −1.02, 2.54 | 0.402 | −0.14 | −2.16, 1.88 | 0.891 |
| VR and GPM+ (Arm 2) *vs.* control (Arm 3) | 0.19 | −2.06, 2.44 | 0.869 | −0.70 | −2.90, 1.50 | 0.533 |
| Intervention effects at 16 weeks | | | | | | |
| VR (Arm 1) *vs.* control (Arm 3) | 1.50 | −0.50, 3.51 | 0.142 | 0.04 | −1.76, 1.85 | 0.962 |
| VR and GPM+ (Arm 2) *vs.* control (Arm 3) | 3.67 | 0.95, 6.38 | 0.008 | 0.90 | −1.16, 2.96 | 0.393 |
| *Mental health literacy* | | | | | | |
| Intervention effects at 8 weeks | | | | | | |
| VR (Arm 1) *vs.* control (Arm 3) | 4.30 | 1.58, 7.02 | 0.002 | 1.75 | −1.23, 4.73 | 0.249 |
| VR and GPM+ (Arm 2) *vs.* control (Arm 3) | 7.35 | 4.67, 10.03 | <0.001 | 3.94 | 0.95, 6.93 | 0.010 |
| Intervention effects at 16 weeks | | | | | | |
| VR (Arm 1) *vs.* control (Arm 3) | 0.73 | −1.76, 3.22 | 0.567 | 0.50 | −2.36, 3.36 | 0.732 |
| VR and GPM+ (Arm 2) *vs.* control (Arm 3) | 3.25 | −0.13, 6.63 | 0.059 | 2.61 | −0.53, 5.76 | 0.103 |
| Mental health stigma | | | | | | |
| Intervention effects at 8 weeks | | | | | | |
| VR (Arm 1) *vs.* control (Arm 3) | −3.10 | −5.97, −0.23 | 0.034 | −0.78 | −3.91, 2.35 | 0.627 |
| VR and GPM+ (Arm 2) *vs.* control (Arm 3) | −2.12 | −4.91, 0.68 | 0.137 | −0.99 | −4.16, 2.18 | 0.539 |
| Intervention effects at 16 weeks | | | | | | |
| VR (Arm 1) *vs.* control (Arm 3) | 3.08 | −0.04, 6.21 | 0.053 | 5.77 | 2.40, 9.13 | 0.001 |
| VR and GPM+ (Arm 2) *vs.* control (Arm 3) | −0.05 | −3.09, 2.99 | 0.976 | 3.17 | 0.22, 6.12 | 0.035 |
| Self-compassion | | | | | | |
| Intervention effects at 8 weeks | | | | | | |
| VR (Arm 1) *vs.* control (Arm 3) | 0.36 | −1.72, 2.44 | 0.732 | 2.58 | 0.38, 4.79 | 0.022 |
| VR and GPM+ (Arm 2) *vs.* control (Arm 3) | 2.62 | 0.57, 4.66 | 0.012 | 1.38 | −0.43, 3.19 | 0.135 |
| Intervention effects at 16 weeks | | | | | | |
| VR (Arm 1) *vs.* control (Arm 3) | 2.39 | 0.12, 4.65 | 0.039 | 3.26 | 0.85, 5.68 | 0.008 |
| VR and GPM+ (Arm 2) *vs.* control (Arm 3) | 3.48 | 1.09, 5.88 | 0.004 | 3.38 | 1.20, 5.55 | 0.002 |
| Mental well-being | | | | | | |
| Intervention effects at 8 weeks | | | | | | |
| VR (Arm 1) *vs.* control (Arm 3) | −2.01 | −4.95, 0.93 | 0.180 | 0.97 | −2.02, 3.96 | 0.525 |
| VR and GPM+ (Arm 2) *vs.* control (Arm 3) | −0.35 | −3.12, 2.42 | 0.805 | −2.50 | −5.01, 0.01 | 0.051 |
| Intervention effects at 16 weeks | | | | | | |
| VR (Arm 1) *vs.* control (Arm 3) | −2.08 | −5.19, 1.03 | 0.189 | −0.09 | −3.22, 3.04 | 0.956 |
| VR and GPM+ (Arm 2) *vs.* control (Arm 3) | 0.63 | −2.51, 3.76 | 0.695 | −0.81 | −3.53, 1.91 | 0.561 |
| Adaptive coping | | | | | | |
| Intervention effects at 8 weeks | | | | | | |
| VR (Arm 1) *vs.* control (Arm 3) | −0.40 | −2.52, 1.71 | 0.709 | −0.78 | −2.65, 1.08 | 0.410 |
| VR and GPM+ (Arm 2) *vs.* control (Arm 3) | −1.84 | −4.13, 0.46 | 0.116 | −2.04 | −4.09, 0.02 | 0.052 |
| Intervention effects at 16 weeks | | | | | | |
| VR (Arm 1) *vs.* control (Arm 3) | 0.87 | −1.21, 2.96 | 0.411 | 0.07 | −1.81, 1.95 | 0.941 |
| VR and GPM+ (Arm 2) *vs.* control (Arm 3) | 0.04 | −2.32, 2.40 | 0.971 | −0.20 | −2.45, 2.05 | 0.862 |

Note: CI, confidence interval; PHQ-9, Patient Health Questionnaire; Intervention effect is estimated as the interaction between intervention arm and time point, calculated using generalized estimating equation linear regression models with an unstructured correlation matrix; aβ*. Adjusted for prespecified covariates (age) and baseline imbalances (place of birth, level of education, food security and relationship status), and baseline outcome scores.

**Table 5.** Sensitivity analyses of the effectiveness of virtual reality interventions on depression and other mental health outcomes among refugee and displaced youth aged 16–27 years in Kampala, Uganda, 2022

| | β | 95% CI | p-value | aβ* | 95% CI | p-value |
|---|---|---|---|---|---|---|
| **Depression (PHQ–9)** | | | | | | |
| No to any intervention | | Ref | | | Ref | |
| Yes, to any intervention | −1.81 | −3.64, 0.02 | 0.05 | −1.93 | −3.20, −0.65 | 0.003 |
| *Mental health literacy* | | | | | | |
| No to any intervention | | Ref | | | Ref | |
| Yes, to any intervention | 2.26 | 0.78, 3.74 | 0.003 | 1.90 | 0.50, 3.29 | 0.008 |
| Mental health stigma | | | | | | |
| No to any intervention | | Ref | | | Ref | |
| Yes, to any intervention | −1.25 | −4.03, 1.53 | 0.379 | −1.03 | −3.95 −1.90 | 0.022 |
| Mental well-being | | | | | | |
| No to any intervention | | Ref | | | Ref | |
| Yes, to any intervention | −0.62 | −1.43, 0.20 | 0.139 | −0.38 | −1.11, 0.36 | 0.317 |
| Self-compassion | | | | | | |
| No to any intervention | | Ref | | | Ref | |
| Yes, to any intervention | 1.69 | 0.53, 2.86 | 0.004 | 1.74 | 0.70, 2.79 | 0.001 |
| Adaptive coping | | | | | | |
| No to any intervention | | Ref | | | Ref | |
| Yes, to any intervention | 0.96 | −0.52, 2.44 | 0.205 | 1.42 | 0.44, 2.39 | 0.004 |

Note: CI, confidence interval; PHQ-9, Patient Health Questionnaire; Intervention effect is estimated as the interaction between intervention arm and time point, calculated using generalized estimating equation linear regression models with an unstructured correlation matrix; aβ*. Adjusted for prespecified covariates (age, gender), baseline imbalances (place of birth, level of education, food security and relationship status) and baseline outcome scores.

interventions are associated with increased self-compassion in high-income settings, with mixed results regarding improving depression (Falconer *et al.* 2016; Halim *et al.* 2023; Hidding *et al.* 2024). We also found improved self-compassion in the VR and GPM+ arm; we did not identify other GPM+ studies that evaluated its impact on self-compassion, so this is an area of future research for both PM+ and GPM+. Our study contributes to this knowledge base of VR, and VR alongside GPM+, as strategies to improve self-compassion in an LMIC context.

We also found increased mental health literacy among the VR and GPM+ arm. As this improvement was not reported in the VR-only arm, it suggests that GPM+ had particular benefits on improving mental health literacy, referring to knowledge and beliefs that help with understanding, preventing and caring for mental health challenges (Jorm 2012). This aligns with the psychoeducation focus of GPM+ (Dawson *et al.* 2015). As mental health literacy is understudied in LMICs at large, including in

African and humanitarian settings (Fox *et al.* 2022; Sodi *et al.* 2022), our findings identify VR alongside GPM+ as a promising approach to improve mental health literacy among youth in an urban LMIC humanitarian setting. As mental health literacy is associated with reduced adolescent psychological distress *via* improved psychological resilience (Zhang *et al.* 2023), and help-seeking self-efficacy (Kutcher *et al.* 2016; Sodi *et al.* 2022) in other settings, future research can explore the benefits of mental health literacy with urban refugee youth in LMICs and apply strategies such as VR and GPM+.

Although improvements in positive mental health outcomes of self-compassion and mental health literacy in our study were corroborated in sensitivity analyses, our primary outcome of depression did not improve with either intervention arm. Our finding that the VR and GPM+ arm was not associated with reduced depression does not align with a study among conflict-affected adults in Nepal that found modest reductions following GPM+ in psychological distress and depression symptoms (Jordans *et al.* 2021). This finding suggests that GPM+ may need to be further tailored for refugee adolescents and youth in Kampala to be efficacious in reducing depression. There is a scant evidence base assessing the effectiveness of GPM+ on reducing depression with youth, so further efficacy research with youth in LMIC conflict-affected settings is needed. By engaging with the interventions, it was expected that participants may be equipped with the information, skills and tools to improve their mental health outcomes, including depression. However, social and structural inequities are associated with pervasive and persistent depression among urban refugee youth in Kampala (Logie *et al.* 2022) – including food insecurity, violence and lower social support. Therefore, it is plausible that interventions such as ours that do not address these larger social-ecological drivers of depression may not be effective. Our findings are also corroborated by the mixed results reported in a review of VR's effectiveness on reducing depression in high-income settings (Rowland *et al.* 2022). A review of mental health and psychosocial support programs on youth health in LMIC humanitarian settings found that among many intervention modalities, only CBT effectively reduced depression symptoms (Bangpan *et al.* 2024); thus, future VR approaches could integrate CBT modalities.

While we found increased depression among Arm 2 at 16 weeks, this was only significant among young men in gender-disaggregated analyses and was not robust to sensitivity analyses. This suggests that the primary analyses of depression results could be influenced by some sociodemographic characteristics and design features (Thabane *et al.* 2013). Social-contextual diversity in young refugee men's experiences across Kampala's informal settlements is evidenced in our findings of baseline differences in depression, age, place of birth, education level and food insecurity between study arms. This reflects the social organization of urban refugees in Kampala that poses challenges in designing RCTs with minimal baseline differences, as noted in past research (Logie *et al.* 2023). It also raises the question of what we miss by controlling for variables such as food insecurity – that are known drivers of depression with this population (Logie *et al.* 2022) – that were higher in particular sites and how better understanding these baseline differences in food insecurity and depression may shed insight into larger social-ecological stressors and "meaningful sources of variation in the population" (p. 2) (Shapiro *et al.* 2021) that shape depression.

One unexpected finding was increased mental health stigma at 16 weeks, which in gender-disaggregated analyses was only significant among young women. Mental health stigma is a barrier to help-seeking among refugee youth (Marshall and Begoray 2019; Nickerson *et al.* 2020), and there is limited evidence of efficacious

mental health stigma reduction approaches with urban refugee youth in LMICs (Mehta *et al.* 2015). There are also mixed findings regarding VR and mental health stigma reduction, with both significant (Yuen and Mak 2021) and nonsignificant effects (Lem *et al.* 2022) in other global contexts. It is plausible that our intervention did not sufficiently address mental health stigma drivers (*e.g.* social norms, values) (Stangl *et al.* 2019), and we did not apply an intersectional approach that addressed the intersection of gender-based stigma and mental health stigma (Logie *et al.* 2024a). Intersectional approaches to mental health stigma reduction (Sievwright *et al.* 2022) are particularly important, as prior work in the United States noted gender, racial and ethnic identity differences in adolescent mental illness stigma and called for tailored stigma reduction (DuPont-Reyes *et al.* 2020). In sensitivity analyses, *any* intervention participation was associated with reduced stigma; thus, our finding of increased stigma was not robust to sensitivity analyses and changed with the inclusion of both intervention arms, signaling the potential role of sociodemographic factors and/or sociocultural differences between study arms in shaping stigma outcomes.

There are several study limitations. First, the baseline differences between study arms were adjusted for in our analyses, yet indicate there may be sociocultural and contextual differences between study sites that can be considered when interpreting study results. Kampala hosts refugees from numerous countries who may live in informal settlements with others from similar communities, in turn creating sociocultural differences between informal settlements. In order to increase study pragmatism we included refugee youth in Kampala from many countries of origin, yet this increased heterogeneity in our sample and study design. We opted to randomize by site, rather than by individual, due to the shared physical and social environment of slums that could introduce contamination between arms (Ezeh *et al.* 2017). Although we accounted for clustering of individual repeated observations, we had insufficient sites to account for clustering within sites. As noted in prior research with this population: "This social organization and socio-cultural diversity of urban refugees in Kampala's informal settlements presents challenges in designing a cluster randomized trial that requires multiple clusters (*e.g.* informal settlements) per condition with minimal baseline differences" (Logie *et al.* 2023) (p. 11) and calls for larger randomized trials with methodological innovation. Second, the nonrandom sample precludes extrapolating findings to all urban refugee youth in Kampala. Third, we did not restrict inclusion criteria based on mental health screening to only include persons with moderate to severe levels of depression, and this may have allowed more accurate evaluations of the effect of the VR interventions on depression.

## Conclusion

Our novel findings contribute to the growing evidence base of VR as a tool for addressing positive mental health outcomes in high-income settings, to show its efficacy in improving self-compassion with urban refugee youth in an LMIC context such as Kampala. Approaches for building resilience and positive psychology outcomes, as well as mental health literacy, with urban refugee youth in Kampala can incorporate VR and GPM+. VR-based interventions such as those implemented in this study involve up-front, one-time costs in developing the VR experience and purchasing VR headsets, but the headsets can be cleaned and reused and the VR experience can be similarly used again, signaling the possibility

of scalability. GPM+, however, requires staff to coordinate logistics and implement in-person sessions, so is time and labor-intensive and would require more long-term costs in scaling up. While the VR approaches we tested were not efficacious in addressing depression, future multilevel approaches can supplement VR to address social and structural inequities that drive depression in urban refugee youth, such as violence, food insecurity and social isolation (Logie *et al.* 2022).

**Open peer review.** To view the open peer review materials for this article, please visit http://doi.org/10.1017/gmh.2025.3.

**Supplementary material.** The supplementary material for this article can be found at http://doi.org/10.1017/gmh.2025.3.

**Data availability statement.** Available upon reasonable request from C. Logie (carmen.logie@utoronto.ca) and upon obtaining required research ethics board approvals in Canada and Uganda.

**Acknowledgments.** We would like to acknowledge the support and contributions of Young African Refugees for Integral Development (YARID), Uganda Ministry of Health, Uganda National AIDS Control Program, Dr. Gabby Serafini (WelTel), Mildmay Uganda, Organization for Gender Empowerment and Rights Advocacy (OGERA Uganda), Most At Risk Population Initiative, Uganda Office of the Prime Minister Department of Refugees and Tushirikiane Peer Navigators.

**Author contribution.** *Study design:* CHL, MO, NK, RH, DKM, PK, LM. Data collection: CHL, MO, LG, JLK, NK, RH, DKM, AN, BK, RL. *Data management:* CHL, ZA, FM, JLK, RH, DKM, AN, BK, PK, RL, LM. *Manuscript writing:* CHL, ZA, FM, LM. *Manuscript editing:* MO, LG, JLK, NK, RH, DKM, AN, BK, PK, RL.

**Financial support.** This study is funded by the Canadian Institutes of Health Research (CIHR) (Project Grant 389142) and Grand Challenges Canada (R-GMH-POC-2107-43740). The funding agencies played no role in the design or execution of the study. CHL is also funded by the Canada Research Chairs program (Tier 2: Logie), Canada Foundation for Innovation (Logie Lab) and the Canadian Institutes of Health Research (COVID-19 Wider Impacts Grant). Gittings was also supported by the Social Sciences and Humanities Research Council of Canada (postdoctoral fellowship).

**Competing interest.** RL is an academic physician-researcher and also has interests in a nonprofit and private company social enterprise, WelTel Inc., that develops and provides digital health software. He is not being paid or otherwise compensated by WelTel for this project. No other authors declare a conflict of interest.

**Ethics statement.** The Tushirikiane-4-MH trial has been approved by the University of Toronto Research Ethics Committee (May 12, 2021), Mildmay Uganda Research Ethics Committee (June 24, 2021) and the Uganda National Science and Technology Council (January 6, 2022). This trial is registered at ClinicalTrials.gov (NCT05187689). All participants provided written informed consent with the support of a peer navigator after receiving information about the study to ensure an understanding of refusal/withdrawal rights, study process and expectations. We received ethical approval to allow teens aged 16–17 to participate without parental consent.

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
