## [Reviewer Report]

This study tackles an important issue but suffers from several critical concerns regarding its research design, methodological approach, and interpretation of intervention effectiveness. Although the use of virtual reality (VR) for improving the mental health of refugee youth holds promise, this study falls short of meeting those expectations, leaving substantial doubts about the intervention’s efficacy and its broader applicability. The research requires significant refinement and further investigation.

1. In low-income countries and humanitarian crisis settings, interventions need to be both simple and cost-effective. VR technology, however, necessitates expensive equipment and technical support, which raises questions about the availability of the necessary infrastructure and resources to sustain such interventions in these environments. The study would benefit from a more thorough discussion of the practical challenges and feasibility of implementing VR interventions within refugee communities in Uganda over the long term.

2. Randomized controlled trials (RCTs) are integral to assessing the efficacy of interventions, but this study demonstrates significant imbalance in baseline depression levels between the groups, undermining the validity of its findings. For example, the control group (Arm 3) had a considerably higher mean depression score of 9.22 compared to the VR intervention group (Arm 1), which had a mean score of 5.62. This discrepancy complicates the interpretation of whether observed changes in the intervention groups were attributable to the intervention itself. Such an imbalance significantly weakens the internal validity of the study. To more accurately evaluate the effect of VR on depression, it would have been more appropriate to include participants with moderate or higher levels of depression at baseline. Given that only 27.5% of participants exhibited moderate to severe depression, the study’s design may not be well-suited for assessing improvements in depression. A more detailed explanation of the exclusion criteria would help address this issue.

3. The VR intervention, a central element of this study, requires further explanation. Specifically, the study needs to clearly articulate the mechanisms through which the intervention was expected to reduce depression. Given that the VR content was focused on improving health literacy, reducing mental health stigma, and promoting self-compassion, the observed changes in these secondary outcomes are not unexpected. However, a more in-depth discussion is needed to explain why the primary outcome—depression—did not show significant change, considering both the research design and the unique characteristics of the study participants.

---

## [Reviewer Report]

Comments to authors:

Overall:

Thank you for the opportunity to review this study. I enjoyed reading it. VR, as a mental health intervention by itself, and the combination of VR and GPM+, are both interesting and novel especially in LMIC settings. My overall feedback is that it would be helpful to clarify why a three-arm approach was chosen for this study. The overall framing of the study and the presentation, especially in the introduction and discussion, focus much more on the VR rather than VR and GPM+. Is the focus of the study to test the added potential benefit of GPM+ to VR compared to VR alone or to test the use of VR in general? The paper is currently written as if the purpose is to test the benefits of VR alone and if that is the case, why include a VR and GPM+ arm? I have some recommendations below to further refine the approach.

Introduction:

- Line 98 – should be forcibly displaced not displayed

- In the last paragraph of the Introduction, it would be helpful to add how the study aims to address the knowledge gaps outlined in the previous paragraph. What makes the VR and GPM+ approach different from the referenced studies? How could research on this intervention help to potentially fill in the gaps that currently exist? What is the evidence base for using VR as a mental health intervention?

- I find that the combination of VR and GPM+ is novel and could be helpful for adolescents. Could you share more information on why these two interventions are being combined? Since arm 1 and 2 are being compared with the SOC, what is the added value of using VR before GPM+?

- I would suggest shifting some of the background information about VR and GPM+ to the introduction and this may be helpful to address some of the points I listed in the previous bullets. The description of the interventions in the methods section could then go into further detail on how the two interventions were implemented.

- It is mentioned in the 2nd sentence of the last paragraph of the Introduction that feasibility is also being evaluated. However, feasibility is not listed in the primary or secondary study objectives that are highlighted at the end of the paragraph starting with the sentence “the primary study objective is...”. How was feasibility measured or determined?

Methods

- I find the section about recruitment from the previous studies to be slightly confusing and I would suggest clarifying some details around how the recruitment occurred. Please clarify what you mean by “additional purposive recruitment” and how exactly participants were recruited from the previous studies. Aside from the inclusion criteria listed, were there any exclusion criteria? Did randomization to the three study arms occur all at once or were participants recruited and randomized at different time points? In the results section, I see that there were statistically significant differences at baseline of some of the primary and secondary outcomes as well as the demographics across arms, so clarifying details in the methods section on recruitment procedures would be helpful to understand these differences.

- Were there any attempts at masking the research team to the study allocation of the participants?

- Was the VR experience (Arm 1) a one-time intervention or did participants receive several sessions?

- Aside from identifying as refugees or displaced persons and their age, what were other criteria for becoming a peer navigator to deliver GPM+? How were they trained and supervised? How were issues of safety related to suicidality managed? Were questions on self-harm asked?

- How were participants divided into various PM+ group? Was it based on location, age, or gender? Rather than providing justification of the use of VR and GPM+ in the Methods section, I would suggest moving this background information to the introduction and focusing the Intervention Approaches section within the Methods on how specifically VR was implemented in Arm 1 and VR and GPM+ was implemented in Arm 2.

- What exactly happens in the 15-minute VR session? For example, is it a virtual psychoeducation session that is pre-recorded? Is it a game? Who is delivering or facilitating the use of VR?

- What was the primary timepoint for measuring outcomes?

Results and Discussion

- Though included in the table, I think it is also important to mention in the text the statistically significant differences at baseline for some of the secondary outcomes.

- Table 5 – this is the first time that the term mHealth is being used to refer to the interventions. I would suggest change this in the title to align with the other tables.

- In the discussion section, the 2nd paragraph focuses on VR and improvements in self-compassion but this outcome was also observed in the VR and GPM+ arm. How does this outcome compare to other GPM+ studies?

- In the discussion paragraph starting with “although improvements...”, while the findings on depression outcomes may be in line with mixed results from other VR studies, there is some evidence for reduction in depression symptoms for GPM+. Please include how findings in this study compare to other GPM+ studies.

- Do you think there are any issues of scalability or implementation around VR based interventions? or VR and GPM+?

- One limitation is also that there was not any inclusion criteria based on mental health assessments.

---

## [Editor Report]

We have received feedback from two reviewers. Both reviewers were enthusiastic about the paper, but acknowledged some limitations and areas to improve the manuscript. We encourage the reviewers to revise the manuscript according to these comments with particular attention to the requests for clarifying certain points and addressing some of the methodological limitations. Thank you for submitting your manuscript to Global Mental Health.